# Effects of Organophosphate-Degrading Bacteria on the Plant Biomass, Active Medicinal Components, and Soil Phosphorus Levels of *Paris polyphylla* var. *yunnanensis*

**DOI:** 10.3390/plants12030631

**Published:** 2023-01-31

**Authors:** Zhuo-Wei Li, Yue-Heng Wang, Chang Liu, Ying-Mei Wu, Guo-Xin Lan, Yan-Bin Xue, Qiang-Sheng Wu, Nong Zhou

**Affiliations:** 1Chongqing Engineering Laboratory of Green Planting and Deep Processing of Famous-Region Drug in the Three Gorges Reservoir Region, College of Biology and Food Engineering, Chongqing Three Gorges University, Chongqing 404120, China; 2College of Environmental and Chemical Engineering, Chongqing Three Gorges University, Chongqing 404120, China; 3College of Horticulture and Gardening, Yangtze University, Jingzhou 434023, China; 4College of Pharmacy, Engineering Center of State Ministry of Education for Standardization of Chinese Medicine Processing, Nanjing University of Chinese Medicine, Nanjing 210023, China

**Keywords:** active ingredients, biomass, organophosphate-degrading bacteria, P, *Paris polyphylla* var. *yunnanensis*

## Abstract

*Paris polyphylla* var. *yunnanensis*, a medicinal plant that originated in Yunnan (China), has been over-harvested in the wild population, resulting in its artificial cultivation. Given the negative environmental impacts of the excessive use of phosphorus (P) fertilization, the application of organophosphate-degrading bacteria (OPDB) is a sustainable approach for improving the P use efficiency in *Paris polyphylla* var. *yunnanensis* production. The present work aimed to analyze the effects of three organic phosphate-solubilizing bacteria of *Bacillus* on the yield and quality of *P. polyphylla* var. *yunnanensis* and the P concentrations in the soil. All the inoculation treatments distinctly increased the rhizome biomass, steroidal, and total saponin concentrations of the rhizomes and the Olsen-P and organic P in the soil. The highest growth rate of rhizomes biomass, steroidal saponins, available phosphorus, and total phosphorus content was seen in the S7 group, which was inoculated with all three OPDB strains, showing increases of 134.58%, 132.56%, 51.64%, and 17.19%, respectively. The highest total saponin content was found in the group inoculated with *B. mycoides* and *B. wiedmannii*, which increased by 33.68%. Moreover, the highest organic P content was seen in the group inoculated with *B. wiedmannii* and *B. proteolyticus,* which increased by 96.20%. In addition, the rhizome biomass was significantly positively correlated with the saponin concentration, together with the positive correlation between the Olsen-P and organic P and total P. It is concluded that inoculation with organophosphate-degrading bacteria improved the biomass and medicinal ingredients of the rhizome in *P. polyphylla* var. *yunnanensis*, coupled with increased soil P fertility, with a mixture of the three bacteria performing best.

## 1. Introduction

*Paris polyphylla* var. *yunnanensis* is a perennial herb in the genus Paris of the Liliaceae family [1]. It is a native Yunnan Chinese herb, which is primarily used as a medicine due to its antitumor, antipyretic, sedative, and analgesic effects [2,3,4,5]. The current sharp increase in market demand for *P. polyphylla* var. *yunnanensis* has led to the over-harvesting of wild plants. The low reproductive rate and slow growth of the rhizomes, the part used medicinally, limits the development of a sustainable pharmaceutical industry [6]. Therefore, artificial cultivation of *P. polyphylla* var. *yunnanensis* has emerged to achieve sustainable harvesting and conserve wild populations. 

As an essential element of plant growth, development, and physiological activities [7], phosphorus can not only enhance the synthesis and transport of carbohydrates, promote protein synthesis, and accelerate fat metabolism, but also enhance the drought, cold, and disease resistance of crops [8]. Soil available phosphorus (Olsen-P) is a form of phosphorus that can be directly absorbed and utilized by plants. It can promote plant growth and affect both soil enzyme activity and microbial content, and it is an important index of soil fertility [9]. Due to immobilization, most P in the soil exists in the form of insoluble compounds that plants are unable to absorb and use directly. Although the application of P fertilizer can alleviate the P deficiency in plants, the application of large amounts of fertilizer can cause a soil nutrient imbalance, reduced fertilizer efficiency, soil slumping, and soil degradation through nutrient loss [10,11]. Therefore, it is urgent to find a suitable fertilizer to replace phosphorus fertilizer, which can not only increase the available value of *P. polyphylla* var. *yunnanensis*, but also protect the ecological environment reasonably.

It is documented that organophosphate-degrading bacteria play an important role in the P acquisition of plants for the regulation of plant growth and development [12,13,14,15]. It has been found that the OPDB in the soil can mineralize organophosphate compounds and convert insoluble compounds into elements that can be directly absorbed and utilized by plants [16], thereby alleviating plant P deficiency. Furthermore, inoculation with OPDB not only increases the P, copper, and iron acquisition in plants [12,13,14], but also promotes the accumulation of biomass [17], thus playing a dual role in enhancing the uptake and utilization of soil P and promoting plant growth [18]. The application effect of organophosphate-degrading bacteria varies for different bacteria strains, plant conditions, and soil types. It was more effective under glasshouse conditions than under field conditions [17]. OPDBs solubilized Ca–P complexes more effectively in calcareous soils than in alfisols [14]. Earlier studies on the interactions between OPDB and plants have focused on wheat [19], maize [20], peanut [21], and other cash crops [22,23], while there are relatively few studies on the OPDB–plant interactions in Chinese herb cultivation. The effects of inoculating *P. polyphylla* var. *yunnanensis* with different OPDB species or complex bacteria have not yet been studied. Therefore, it is important to explore the use of organophosphorus fertilizers to obtain high-yielding, high-quality herbs during the artificial cultivation of *P. polyphylla* var. *yunnanensis*.

In this study, *P. polyphylla* var. *yunnanensis* seedlings were inoculated with various combinations of three dominant OPDB [24], and their effects on the yield and quality of the *P. polyphylla* var. *yunnanensis* were measured through the biomass of the rhizomes, the concentration of medicinal compounds, and the changes in the P forms in the soil, with the aim of providing a theoretical basis for the development and application of biofertilizers for *Paris polyphylla* var. *yunnanensis*.

## 2. Results

### 2.1. Changes in Rhizome Biomass

Inoculation with OPDB significantly (*p* < 0.05) increased the rhizome biomass of *P. polyphylla* var. *yunnanensis*, with the greatest increases being seen in the S7 (Sample 7), S6 (Sample 6), and S3 (Sample 3) groups (134.58%, 105.60%, and 102.83%, respectively) compared to the CK group. The largest increase was seen in group S7, with a mixture of all three bacteria. No significant differences were found in the drying rates of the rhizomes (Figure 1).

### 2.2. Changes in Steroidal Saponin Content

#### 2.2.1. Changes in Steroidal Saponin Concentrations in the Rhizome

The changes in the steroidal saponin and total saponin concentrations in the rhizomes of *P. polyphylla* var. *yunnanensis* after inoculation with OPDB are shown in Figure 2. The concentrations (mg·g^−1^) of pseudoprotodioscin, polyphyllin VII, polyphyllin II, and polyphyllin I in the treated groups were significantly higher (*p* < 0.05) than in the CK group. Polyphyllin H was significantly increased in all the treatment groups, except S2, and dioscin was increased in all the treatment groups, except S4 and S6. Among them, the pseudoprotodioscin, polyphyllin H, and dioscin concentrations in group S5 showed the greatest increases (152.17%, 87.78%, and 159.46%, respectively). Moreover, polyphyllin VII in group S3, polyphyllin II in group S7, and polyphyllin I in group S1 showed the greatest increases (114.16%, 168.27%, and 245.70%, respectively). The sum of the steroidal saponin concentrations showed that the treatment groups were all significantly higher than the CK group. The highest growth rate was 132.56% in the S7 group, followed by the S3 (118.10%) and S1 (109.42%) groups. The total saponin concentration in the rhizomes was significantly increased by the inoculation treatments, with the highest growth rate (33.68%) observed in the S4 group, followed by the S7 group. Overall, inoculation with all three treatments had the greatest effect on the saponin concentrations.

#### 2.2.2. Principal Component Analysis of the Six Steroidal Saponins

The results of the principal component evaluation are presented in Table 1. Two principal components with eigenvalues greater than 1 were extracted. Principal component 1, with an eigenvalue of 3.221 and a contribution margin of 53.68, mostly reflected information on polyphyllin H and polyphyllin I. The eigenvalue of principal component 2 was 1.940, with a contribution margin of 32.34, reflecting information on polyphyllin II and polyphyllin VII. 

The principal component scores were calculated and ranked for the different inoculation treatments, as shown in Table 2. In F1 (principal component 1), the highest score was seen in the S5 treatment group, indicating that the S5 group was the most abundant in polyphyllin H and polyphyllin I. In F2 (principal component 2), the highest score was observed in the S3 treatment group, indicating that the S3 group was the most abundant in Polyphyllin II and Polyphyllin VII. The highest overall score was seen in the S7 treatment group. On the whole, the S7 treatment group was the most abundant in the six steroidal saponins of *P. polyphylla* var. *Yunnanensis*. This analysis should aid in choosing the best treatment.

### 2.3. Changes in Concentrations of Different Soil P Forms

The changes in the total P, Olsen-P, and organic P content in the rhizospheric soil of *P. polyphylla* var. *yunnanensis* after OPDB inoculation are shown in Figure 3. The inoculation treatment significantly increased the concentrations of the total P, Olsen-P, and organic P in the rhizospheric soil, with the highest increase in the total P and Olsen-P concentrations being seen in group S7, with growth rates of 17.19% and 51.64%, respectively, and the highest increase in the organic P concentrations being in group S6, with a growth rate of 96.20%.

In Figure 4, the ratios of the Olsen-P and organic P to the total P indicate that each treatment group had a higher percentage of P than that observed in the non-inoculated CK group. The highest percentage of Olsen-P was seen in the S7 group (16.61%) and the highest percentage of organic P (15.99%) was seen in S6, although S7 had the greatest overall effect. Inoculation with OPDB significantly increased the total P, Olsen-P, and organic P concentrations in the rhizospheric soil of *P. polyphylla* var. *yunnanensis*, enhancing the conversion capacity of the Olsen-P and thus improving the fertility of the rhizospheric soil.

### 2.4. Correlation Analysis

The results of the correlation analysis of the biomass, steroidal saponin, and total saponin in the rhizomes of *P. polyphylla* var. *yunnanensis*, as well as the organic P, Olsen-P, and total P concentrations of the rhizospheric soil, are presented in Table 3. Biomass was significantly positively correlated (*p* < 0.05) with the polyphyllin VII and Olsen-P content and highly significantly correlated (*p* < 0.01) with the polyphyllin II content; pseudoprotodioscin was significantly correlated with the polyphyllin H and dioscin content; and the Olsen-P and total P concentrations showed highly significant positive correlation.

## 3. Discussion

The rhizome of *P. polyphylla* var. *yunnanensis* is rich in steroidal saponins, and a quantitative analysis of the rhizome biomass in *P. polyphylla* var. *yunnanensis* can indirectly reveal the yield of the medicinal materials of *P. polyphylla* var. *yunnanensis* [25]. The active medicinal ingredients in the rhizomes of *P. polyphylla* var. *yunnanensis* are primarily steroidal saponins that have anti-inflammatory and anti-tumor effects [26,27,28,29,30,31]. The 2020 edition of the *Chinese Pharmacopoeia* includes polyphyllin I, II, and VII as quality control indicators [2], and studies have also shown that polyphyllin H has hemostatic effects [32], dioscin has strong immunomodulatory effects [33,34,35], and pseudoprotodioscin can reduce the gene expression regarding the synthesis of cholesterol and triglycerides [36]. In addition, Gao et al. [37] showed that the saponins of polyphyllin can scavenge reactive oxygen species and significantly inhibit lipid peroxidation. In this study, the effects of inoculation with different OPDB on the biomass, steroidal saponin concentration, and total saponin concentration of the rhizomes of *P. polyphylla* var. *yunnanensis* varied, although each treated group had significantly better results than the non-inoculated CK group, supporting the hypothesis that inoculation with OPDB could improve the utilization of soil P by *P. polyphylla* var. *yunnanensis*. The S7 treatment group with all the three OPDB showed the most significant effects. The results of the correlation analysis showed that the biomass, steroidal saponin, and total saponin content were significantly correlated with each other, indicating that the biomass and saponin concentration in the rhizomes interact to improve the yield and quality of the medicinal components of *P. polyphylla* var. *yunnanensis*. The results of the principal component analysis of the six steroidal saponins showed that the score coefficients of saponins H, I, and II were high; therefore, these three saponins could be used as quality control indices for *P. polyphylla* var. *yunnanensis*, which is consistent with the findings of Gu et al. [38]. The principal component analysis showed that the six steroidal saponins in the rhizomes were most abundant in the S7 group, which is consistent with the results of studies on the biomass and total saponin content of the rhizomes, indicating again that the simultaneous inoculation of the three bacteria had the best effect. 

The rhizosphere phosphorus-solubilizing bacteria can release soluble phosphorus from insoluble phosphate, meaning that they play an important role in the soil phosphorus cycle [39]. The soil P activation coefficient (PAC) is the proportion of Olsen-P to total P, which can reflect the degree of conversion of the total P to Olsen-P [40,41]. In this study, the organic P, Olsen-P, and total P amounts in the treated groups were significantly higher than in the CK group, and the PAC was significantly increased. In the rhizospheric soil, the Olsen-P and organic P concentrations showed significant positive correlation, which is consistent with the results of Qi et al. [40]. Furthermore, the Olsen-P and total P concentrations showed highly significant positive correlation, which is consistent with the results of Wang et al. [42], indicating that the organic P, Olsen-P, and total P contents can interact with each other to improve soil fertility. The effects of different organophosphate-degrading bacteria on the quality of *P. polyphylla* var. *yunnanensis* and the soil P content were discussed in this study. Zhao et al. [43] discussed the effects of different potassium-solubilizing bacteria on the medicinal quality and soil potassium form of *P. polyphylla* var. *yunnanensis*. Huang et al. [44] studied the effects of arbuscular mycorrhizal fungi on the growth and development of *P. polyphylla* var. *yunnanensis* and the content of steroidal saponins. Liu et al. [45] showed that the biotransformation of endophytic fungi could improve the content of saponins and its anti-tumor effect. It can be seen that the growth-promoting effect of microbial fertilizer on *P. polyphylla* var. *yunnanensis* is a hot topic at present, although the related studies have mainly focused on the effects of single species of bacteria. Whether the interaction of different bacteria groups strengthens or weakens the growth-promoting effect on the *P. polyphylla* var. *yunnanensis* is still unknown, meaning that further research and discussion are needed. 

## 4. Materials and Methods

### 4.1. Bacterial Materials

The bacteria strains utilized in the experiment were strains preserved in our own laboratory, which were isolated from the rhizospheric soil of *P. polyphylla* var. *yunnanensis* in the field and identified as *Bacillus mycoides*, *B. wiedmannii*, and *B. proteolyticus* by Du et al. [24]. The bacteria were activated and placed in the solution, and they were then transferred by pipette into a triangular flask containing 50 mL of beef paste peptone medium. Approximately 4 L of each bacterial strain was cultured in this way.

### 4.2. Plant Materials

All the tested seedlings were obtained from the same batch of four-year-old *P. polyphylla* var. *yunnanensis* seedlings with the same size and free from diseases and pests. They were provided by the standardized planting garden in Pulang, Suyang, Yongping, Dali, Yunnan (25°39′13.56” N,99°33′58.38” E), China, and identified. The seedlings were preserved as single plants and conventionally treated to ensure the stability and homogeneity of the germplasm resources. Their substrate was made of common loam, sand, and organic fertilizer at a ratio of 2:1:1, and it was passed through a 2 mm sieve, autoclaved at 121 °C for 30 min, and then cooled and sealed for use.

The potted experiment was carried out at Anshun College, Anshun, Guizhou, China (26°14′37.86′′ N,105°54′3.75′′ E). The seedlings were planted in December 2020. The experiment was divided into eight groups, including seven treatment groups (S1–S7) and one control group (CK). Details of the inoculated treatments are shown in Table 4.

Therefore, a total of 8 treatments were included in this experiment, with each treatment being replicated 10 times and each pot containing 5 seedlings. The inoculated treatment was supplied with 150 mL suspension of designed OPDB per pot. The rhizomes and rhizospheric soil of the seedlings were harvested after they had ripened. After determination of the rhizome biomass, the rhizomes were crushed, sieved through 0.18 mm, sealed, and dried for storage. The rhizospheric soil was obtained by shaking the roots, dried naturally at 25 ± 1 °C, sieved, and sealed for drying and storage.

### 4.3. Determination of Rhizome Biomass

The freshly harvested rhizomes were cleaned before determining their fresh biomass. They were then dried in an electric thermostatic blast oven at 45 °C to a constant weight for the dried biomass. Each treatment group was replicated five times. The drying rate = the dry weight of the rhizome/fresh weight of rhizome [46].

### 4.4. Determination of Steroidal Saponin Concentration in Rhizomes

The content of total saponins was determined using an optimized UV spectrophotometry (Shanghai Jing Hua Ltd., Shanghai, China) [47]. The concentrations of the six steroidal saponins (pseudoprotodioscin, polyphyllin VII, polyphyllin H, polyphyllin II, dioscin, and polyphyllin I) in the rhizomes were determined by means of H-Class ultra-performance liquid chromatography (Waters Inc., Milford, MA, USA). First, 0.5 g of rhizome powder (0.5 g) was incubated with 15 mL of methanol under ultrasonic conditions (SB-5200DTN, Ningbo Xinzhi Biological Technology Co. Ltd., Zhejiang, China, 40 kHz) for 30 min. The extracts were centrifuged at 4000 r/min and the supernatant was filtered through a 0.22 µm microporous membrane for 10 min [38]. Each set of samples was replicated three times. The condition of chromatography was as follows: chromatographic column: Accucore PFP (2.1 × 100 mm, 2.6 µm); mobile phase: acetonitrile (A)–water (B), gradient elution: 0–5 min, 20–25% A; 5–15 min, 25–50% A; 15–17 min, 50–20% A; 17–20 min, 20% A; detection wavelength: 203 nm; flow rate: 0.3 mL/min; injection volume: 5 µL; column temperature: 30 °C ; and sample chamber temperature: 10 °C . The measured peak areas were substituted into the linear regression equation to calculate the concentrations of the various steroidal saponins in the *P. polyphylla* var. *yunnanensis* root samples. The chromatograms of the controls and samples are shown in Figure 5. 

### 4.5. Methodological Validation

The six steroidal saponins had good linearity in the corresponding linear ranges. The precision investigation revealed that the RSDs (*n* = 6) of the peak areas of the pseudoprotodioscin, polyphyllin VII, polyphyllin H, polyphyllin II, dioscin, and polyphyllin I were 0.34%, 1.02%, 1.18%, 1.25%, 0.60%, and 0.33%, respectively, all of which were less than 2.00%, indicating that the precision of the instrument was suitable. Likewise, the repeatability study resulted in good repeatability scores for the assay: I RSDs (*n* = 6) of the peak areas of the pseudoprotodioscin, polyphyllin VII, polyphyllin H, polyphyllin II, dioscin, and polyphyllin I were 1.87%, 0.51%, 2.04%, 0.75%, 0.64%, and 1.63%, respectively, all of which were less than 3.00% (Appendix A). The mean recoveries of the six steroidal saponins ranged from 95.80% to 102.52%, with RSDs between 0.84% and 2.76%, all of which were less than 3.00%, indicating that the assay used in this experiment had high recovery rates as well as accurate and reliable results (Appendix A).

### 4.6. Principal Component Analysis

The concentrations of the six steroidal saponins in the rhizomes of *P. polyphylla* var. *yunnanensis* were used as indicators for the principal component analysis. The Kaiser–Meyer–Olkin (KMO) measure and Bartlett’s sphericity tests resulted in a KMO of 0.270 and a Bartlett’s sphericity chi-square of 41.388, with 15 degrees of freedom and *p* < 0.05. The data were determined to be suitable for the factor analysis.

The principal component scores were calculated with the formulae as follows:*F*_1_ = 0.417*X*_1_ + 0.282*X*_2_ − 0.523*X*_3_ + 0.351*X*_4_ + 0.356*X*_5_ + 0.471*X*_6_
*F*_2_ = −0.458*X*_1_ + 0.513*X*_2_ − 0.079*X*_3_ + 0.541*X*_4_ − 0.459*X*_5_ + 0.130*X*_6_
*F**_Z_* = 0.624*F*_1_ + 0.376*F*_2_

*F*_1_ and *F*_2_ denote the 1st and 2nd principal component scores, respectively, while *X*_1_ to *X*_6_ denote the standardized data of the six factors. The ratio of the characteristic and cumulative contributions corresponding to the two principal components was then used as the weight to calculate and rank the composite score to obtain *F*_Z_ [27].

### 4.7. Determinations of Concentrations of Different P Forms in Rhizospheric Soil

The content of Olsen-P in the rhizospheric soil was determined using the sodium bicarbonate leaching-colorimetric method. The organic P content was determined using the method described by Bao [48]. The total P concentration was determined via the microwave digestion of the samples and then via an FIA-6100 Automatic Flow Injection Analyser (Beijing Jitian Instruments Co. Ltd., Beijing, China).

### 4.8. Statistical Analyses

Microsoft Excel 2010 was used for the data processing and graphing. SPSS 22.0 was used for the principal component analysis and correlation analysis. Adobe Photoshop CC 2019 was used to edit the figures.

## 5. Conclusions

In conclusion, inoculation with different OPDB could effectively increase the Olsen-P content in the rhizospheric soil, thereby promoting the absorption and utilization of phosphorus by *P. polyphylla* var. *yunnanensis* herbs. Moreover, it also increased the biomass and active medicinal components of *P. polyphylla* var. *yunnanensis*. Among them, inoculation with the three OPDB species resulted in the highest content of biomass, steroidal saponins, and Olsen-P. Thus, the inoculation method can be considered in the future artificial cultivation of *P. polyphylla* var. *yunnanensis* as a potent biostimulator. 

## Figures and Tables

**Figure 1 plants-12-00631-f001:**
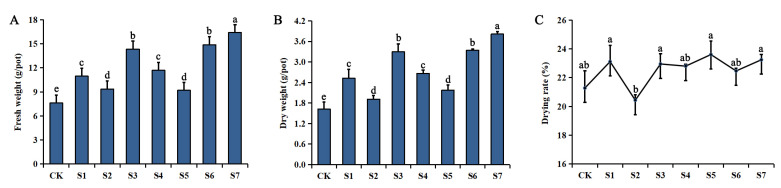
Effects of inoculation with organophosphate-degrading bacteria (OPDB) on the fresh weight (**A**), dry weight (**B**), and drying rate (**C**) of rhizomes of *Paris polyphylla* var. *yunnanensis*. Data (means ± SD, *n* = 3) followed by different letters above the bars indicate significant (*p* < 0.05) differences between treatments. Note: S1–S7 stand for Sample 1–Sample 7.

**Figure 2 plants-12-00631-f002:**
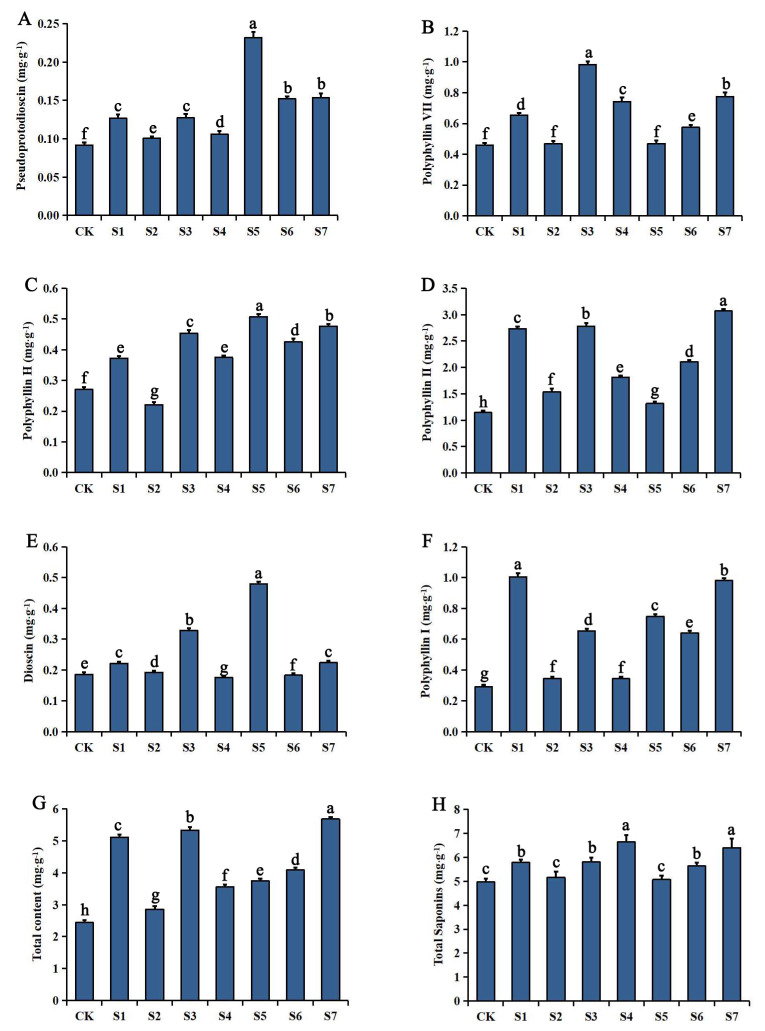
Effects of inoculation with OPDB on the steroidal saponin and total saponin concentrations of *Paris polyphylla* var. *yunnanensis*. (**A**) pseudoprotodioscin; (**B**) polyphyllin VII; (**C**) polyphyllin H; (**D**) polyphyllin II; (**E**) dioscin; (**F**) polyphyllin I; (**G**) total content; and (**H**) total saponins. Data (means ± SD, *n* = 3) followed by different letters above the bars indicate significant (*p* < 0.05) differences between treatments. Note: S1–S7 stand for Sample 1–Sample 7.

**Figure 3 plants-12-00631-f003:**
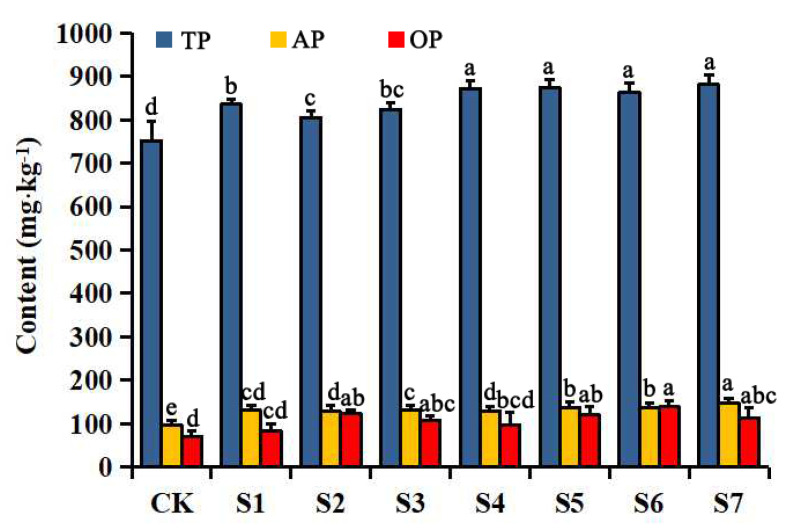
Total P (TP), Olsen-P (AP), and organic P (OP) concentrations in the rhizospheric soils of *P. polyphylla* var. *yunnanensis*. Data (means ± SD, *n* = 3) followed by different letters above the bars indicate significant (*p* < 0.05) differences between treatments. Note: S1–S7 stand for Sample 1–Sample 7.

**Figure 4 plants-12-00631-f004:**
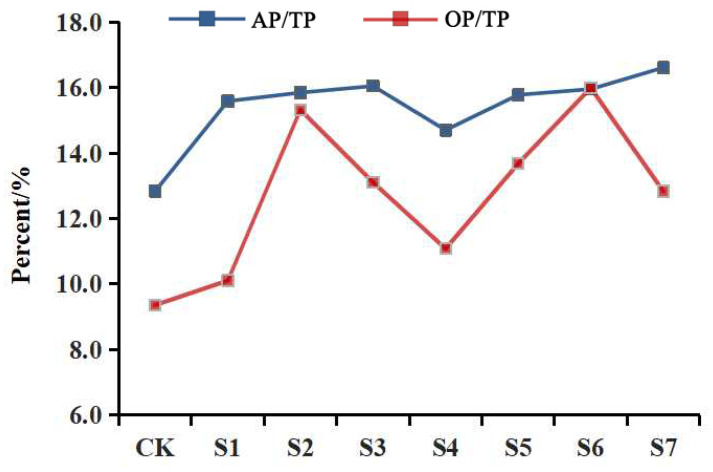
Olsen-P and organic P as a proportion of the total P. Note: S1–S7 stand for Sample 1–Sample 7.

**Figure 5 plants-12-00631-f005:**
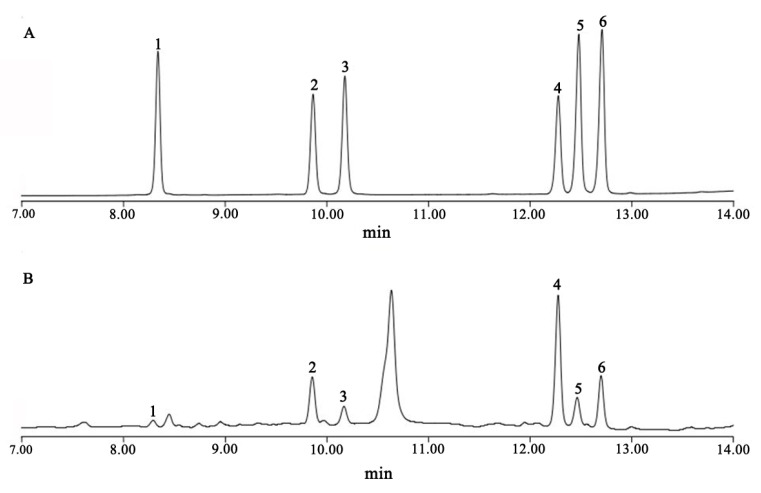
Ultra-performance liquid chromatography diagram of the steroidal saponins of *Paris polyphylla* var. *yunnanensis*. (**A**) Control chromatogram; (**B**) *Paris polyphylla* var. *yunnanensis* chromatogram: 1. pseudoprotodioscin; 2. polyphyllin VII; 3. polyphyllin H; 4. polyphyllin II; 5. dioscin; and 6. polyphyllin I.

**Table 1 plants-12-00631-t001:** Matrix of the principal component analysis of the steroidal saponins of *P. polyphylla* var. *yunnanensis*.

Steroidal Saponins	Principal Component 1	Principal Component 2
Pseudoprotodioscin	0.748	−0.638
Polyphyllin VII	0.507	0.714
Polyphyllin H	0.939	−0.110
Polyphyllin II	0.630	0.754
Dioscin	0.639	−0.640
Polyphyllin I	0.845	0.181
Eigenvalue	3.221	1.940
Contribution rate (%)	53.68	32.34
Cumulative contribution rate (%)	53.68	86.02

**Table 2 plants-12-00631-t002:** Steroidal saponin principal component score of *P. polyphylla* var. *Yunnanensis*.

Treatment Group	F1	Sorting	F2	Sorting	Fz	Sorting
CK	−2.515	8	−0.514	7	−1.763	8
S1	0.716	4	0.940	3	0.800	3
S2	−2.381	7	−0.256	6	−1.582	7
S3	1.439	3	1.183	1	1.343	2
S4	−1.033	6	0.601	4	−0.419	6
S5	1.881	1	−3.048	8	0.027	4
S6	0.070	5	−0.049	5	0.025	5
S7	1.824	2	1.143	2	1.568	1

Note: S1–S7 stand for Sample 1–Sample 7.

**Table 3 plants-12-00631-t003:** Correlation analysis of the biomass, saponin, and P in *Paris polyphylla* var. *yunnanensis*.

Indicators	Biomass	Pseudoprotodioscin	Polyphyllin VII	Polyphyllin H	Polyphyllin II	Dioscin	Polyphyllin I	Total Saponins	Organic P	Olsen-P	Total P
Biomass	1.000										
Pseudoprotodioscin	0.219	1.000									
Polyphyllin VII	0.732 *	−0.123	1.000								
Polyphyllin H	0.673	0.788 *	0.461	1.000							
Polyphyllin II	0.837 **	0.005	0.788 *	0.465	1.000						
Dioscin	−0.044	0.821 *	0.015	0.628	−0.104	1.000					
Polyphyllin I	0.584	0.539	0.331	0.672	0.742 *	0.324	1.000				
Total Saponins	0.702	−0.142	0.709 *	0.361	0.649	−0.319	0.279	1.000			
Organic P	0.437	0.487	−0.018	0.345	0.100	0.218	0.141	0.008	1.000		
Effective P	0.737 *	0598	0.388	0.695	0.600	0.316	0.666	0.481	0.721 *	1.000	
Total P	0.655	0.627	0.314	0.754 *	0.418	0.284	0.540	0.623	0.558	0.890 **	1.000

* indicates significant correlation between the two (*p* < 0.05). ** indicates very significant correlation between the two (*p* < 0.01).

**Table 4 plants-12-00631-t004:** The arrangement of each treatment in this experiment.

Treatments	Details
S1	Inoculation with *Bacillus mycoides*
S2	Inoculation with *B*. *wiedmannii*
S3	Inoculation with *B*. *proteolyticus*
S4	Inoculation with *B. mycoides* and *B. wiedmannii*
S5	Inoculation with *B. mycoides* and *B. proteolyticus*
S6	Inoculation with *B. wiedmannii* and *B. proteolyticus*
S7	Inoculation with *B. mycoides*, *B. wiedmannii*, and *B. proteolyticus*
CK	No inoculation with any organophosphate-degradation bacteria

Note: S1–S7 stand for Sample 1–Sample 7.

## Data Availability

All the data supporting the findings of this study are included in this article.

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
