# Peer review of "Effects of Organophosphate-Degrading Bacteria on the Plant Biomass, Active Medicinal Components, and Soil Phosphorus Levels of Paris polyphylla var. yunnanensis"

_plants, 2023, doi:10.3390/plants12030631_

Round 1

Reviewer 1 Report

This paper studies Effects of Organophosphate-degradation Bacteria on Plant Biomass, Active Medicinal Components, and Soil Phosphorus Levels of Paris polyphylla var. yunnanensis, which has certain theoretical and industrial significance, fluent language and real data. There are still some problems in the writing of the paper, as follows:

 1.      It is suggested to add some specific data in the abstract was.

 2.      57 lines of peanut, initial lowercase

 3.      71-82 lines, S1, S2, S3.... S7 appear for the first time. It is suggested that the text should be written in full, and remarks should be made in the title of Figure 1.

 4.      Lines 83-118 "2.2.1 The content of "Methodological validation" is recommended to be placed in the material method or attachment.

 5.      122 lines, mg ∙ g-1 needs superscript

 6.      S1, S2, S3.... S7 should be noted in the title of all the figures and tables.

 7.      143-193 “2.2.3. The content of "Principal component analysis of the six sterial saponins" is somewhat scattered and does not closely focus on the theme. Suggest rewriting.

 8.      211, 218, 244, 250, 252 genera and species need italics.

9.      Table 6 T1, T2, T3.... T7 corresponds to S1, S2, S3.... S7?

10.    370-371, 391-392, 400, 412, Latin requires italic.

Reviewer 3 Report

In this study, the authors have provided important information for potential Organophosphate Degradation Bacteria on Plant Biomass, Active Medicinal Components, and Soil Phosphorus Levels of Paris polyphylla var. yunnanensis, which would help to enhance the Soil P and fertility. The manuscript is organized and well-written. However, it can be further improved; I have provided a few corrections below. The manuscript requires significant revisions, and then accept it.

1.      A short study gap must be expressed (add) in the abstract between lines 17 to 18. Otherwise, it will not be easy to understand by the audience.  

2.      The study objective in the abstract is too long and ambiguous to understand. Please modify it, Lines 17 to 21.

3.      Introduction section, what kind of research is needed for the time being? Please mention by the end of the paragraph, Line 48.

4.      In the section on materials and method, the author didn’t mention any methodology, how they isolated the bacteria, the protocol and what kind of primers he used for identifying Bacillus strains. Please add the new section with the name of (Isolation and Identification of Bacterial strains). This section should be added before section 4.1. Bacterial materials.

5.      The result section is expressed as per audience demand—no need to change.

6.      Discussion section: I am not satisfied with the first paragraph of the discussion, lines 228 to 247. The authors have written this portion like the introduction. He has to justify his initial results with the latest references. For better understanding, I am attaching the paper for his help; he needs to read and try to construct the story this way.

Asghar, Waleed, and Ryota Kataoka. "Green manure incorporation accelerates enzyme activity, plant growth, and changes in the fungal community of soil." Archives of Microbiology 204.1 (2022): 1-10.

Round 2

Reviewer 2 Report

The authors have successfully revised the manuscript. The comments and suggestions of mine have been well addressed. Current version reads well in high quality. I recommend it accepting for publication as its present form.